# Genome-Wide Association Study Identifies Rice Panicle Blast-Resistant Gene *Pb4* Encoding a Wall-Associated Kinase

**DOI:** 10.3390/ijms25020830

**Published:** 2024-01-09

**Authors:** Yunxin Fan, Lu Ma, Xiaoqian Pan, Pujiang Tian, Wei Wang, Kunquan Liu, Ziwei Xiong, Changqing Li, Zhixue Wang, Jianfei Wang, Hongsheng Zhang, Yongmei Bao

**Affiliations:** State Key Laboratory of Crop Genetics and Germplasm Enhancement, College of Agriculture, Jiangsu Collaborative Innovation Center for Modern Crop Production, Nanjing Agricultural University, Nanjing 210095, China15966697937@163.com (X.P.); 2021101060@stu.njau.edu.cn (P.T.); 2023201042@stu.njau.edu.cn (C.L.); hszhang@njau.edu.cn (H.Z.)

**Keywords:** rice blast, genome-wide association study, Pb4, WAK

## Abstract

Rice blast is one of the most devastating diseases, causing a significant reduction in global rice production. Developing and utilizing resistant varieties has proven to be the most efficient and cost-effective approach to control blasts. However, due to environmental pressure and intense pathogenic selection, resistance has rapidly broken down, and more durable resistance genes are being discovered. In this paper, a novel wall-associated kinase (WAK) gene, *Pb4*, which confers resistance to rice blast, was identified through a genome-wide association study (GWAS) utilizing 249 rice accessions. Pb4 comprises an N-terminal signal peptide, extracellular GUB domain, EGF domain, EGF-Ca^2+^ domain, and intracellular Ser/Thr protein kinase domain. The extracellular domain (GUB domain, EGF domain, and EGF-Ca^2+^ domain) of Pb4 can interact with the extracellular domain of CEBiP. Additionally, its expression is induced by chitin and polygalacturonic acid. Furthermore, transgenic plants overexpressing *Pb4* enhance resistance to rice blast. In summary, this study identified a novel rice blast-resistant gene, *Pb4*, and provides a theoretical basis for understanding the role of WAKs in mediating rice resistance against rice blast disease.

## 1. Introduction

Rice blast, caused by the pathogen *Magnaporthe oryzae* (*M. oryzae*), is one of the most devastating diseases that occurs throughout all growth stages of rice plants and seriously affects both the yield and grain quality of rice crops [1]. Using resistance genes from disease-resistant varieties for breeders is a more cost-effective and environmentally friendly approach. However, most resistance (R) genes conform to the gene-for-gene theory and exhibit race-specific resistance. Due to the high variability of rice blast fungus, the majority of resistance genes will become ineffective within 3–5 years [2]. Thus, more R genes and defense regulator (DR) genes that confer broad-spectrum resistance should be discovered.

To date, over 100 genes conferring resistance to rice blast have been identified, and at least 30 genes have been cloned in rice [2]. However, the majority of these genes are resistant to seedling blast, and only a few, such as *Pb1*, *Pb2*, *Pb3*, *Pi25*, *Pi64*, and *Pi68(t)*, confer resistance to panicle blast [3,4,5,6,7,8]. Among the cloned R genes, the majority encode a nucleotide-binding leucine-rich repeat (NLR) protein, except for *bsr-d1*, *bsr-k1*, *rod1*, *pi21,* and *Pi-d2*, where bsr-d1 functions as a C_2_H_2_-type transcription factor, while *Bsr-k1* encodes a protein containing tetratricopeptide repeats (TPRs) [9]. ROD1 acts as a calcium ion (Ca^2+^) sensor and plays a crucial role in promoting the scavenging of reactive oxygen species (ROS) [10]. *Pi21* encodes a proline-rich protein containing a metal-binding domain [11]. *Pi-d2* encodes a receptor-like kinase protein with a predicted extracellular domain containing the bulb-type mannose-specific binding lectin (B-lectin), as well as an intracellular serine-threonine kinase domain [12]. There are also some receptor-like kinase (RLK) genes conferring resistance to other pathogens, such as *Xa21* and *Xa26,* both encoding an extracellular leucine-rich repeat (LRR) and intracellular kinase domain, conferring resistance to *Xanthomonas oryzae* pv. *oryzae* [13]. Further, *Xa4*, encoding a wall-associated kinase (WAK) protein, provides durable resistance against bacterial leaf blight [14].

WAKs are closely involved in the plant immune process, functioning as receptors of damage-associated molecular patterns (DAMPs). WAKs participate in various physiological processes, including responses to pathogens and metal stress and the regulation of cell elongation. In the context of pathogen response, WAKs are involved in recognizing and transmitting signals from the cell wall to the intracellular components, thereby activating defense responses. They can detect changes in the cell wall caused by pathogen invasion or other environmental stresses, initiating downstream signaling pathways that regulate the expression of defense-related genes and the production of defense compounds [15]. Structurally, WAKs consist of several conserved domains: an amino-terminal (N-terminal) signal peptide, an extracellular WAK galacturonan-binding (GUB) domain, an epidermal growth factor (EGF) domain, an EGF-Ca^2+^ domain, and a carboxy-terminal (C-terminal) cytoplasmic Ser/Thr protein kinase domain [16]. In *Arabidopsis*, the AtGRP3 protein, pectin, and oligogalacturonides (OGs) can bind to the extracellular domain of AtWAK1 and AtWAK2 [17,18,19,20]. WAKs play a crucial role in the plant defense response against pathogens. The transcription factor PIBP1 interacts with PigmR and activates the expression of *OsWAK14*, thereby enhancing the PigmR-mediated defense against rice blast fungus [21]. *OsWAK25* is induced by salicylic acid (SA) and mechanical injury, and overexpression of *OsWAK25* enhances rice blast resistance [22]. The resistance gene *Xa4* provides durable resistance against bacterial leaf blight by strengthening the cell wall by promoting cellulose synthesis and inhibiting cell wall loosening [14].

Genome-wide association studies (GWAS) have become a widely used tool for mapping resistance loci, analyzing the association between a specific resistance phenotype and natural genetic variations, such as single-nucleotide polymorphisms, indels, or the copy number of variations. GWAS is a highly efficient approach that conducts whole-genome scans to identify genomic regions linked to the phenotype of interest. This method has greatly benefited from the rapid advancements in high-throughput sequencing technologies, such as second-generation sequencing, SMART sequencing, and nanopore sequencing [23]. To date, researchers have gained valuable loci through GWAS assays by analyzing the genetic variation, population structure, and diversity in various crop species. For instance, 12 loci associated with bacterial blight resistance were identified through GWAS within the 3000 Rice Genomes Project [24]. A total of 27 rice blast-resistant loci were identified through the GWAS using the Rice Diversity Panel II (C-RDP-II). The rice blast resistance genes *PiPR1*, *RNG1*, and *RNG2* were successfully cloned through GWAS [25,26].

In this study, we conducted a GWAS involving 249 rice cultivars to analyze blast resistance loci and *Pb4* was identified to positively regulate rice resistance. Pb4 is a typical WAK protein, which consists of an N-terminal signal peptide, extracellular GUB domain, EGF domain, EGF-Ca^2+^ domain, TM domain, and intracellular Ser/Thr protein kinase domain. The extracellular domain of Pb4 can interact with the extracellular domain of CEBiP, and its expression is induced by chitin and polygalacturonic acid. These findings provide a theoretical basis for understanding the role of WAKs in mediating rice resistance against rice blast disease.

## 2. Results

### 2.1. Structure and Phenotype of the Population

To map the rice blast-resistant genes, 249 cultivars with relatively uniform growth periods were selected from the 3K rice accessions. The population structure was analyzed using the 3K core SNP, with k (the number of groups) ranging from 2 to 12. The cross-validation error was minimized when k = 7 (Figure 1A,B). Therefore, 249 rice varieties were divided into seven subgroups: 26 GJ-adm, 12 GJ-sbtrp, 30 GJ-tmp, 32 GJ-trpA, 20 GJ-trpB, 32 XI-1A, 49 XI-1B, 20 XI-adm, 14 cA (Aus), and 14 admix individuals (Figure 1C). A maximum likelihood (ML) phylogenetic tree was constructed using the 3K core SNP, showing that individuals with the same group structure were more likely to aggregate together and form a clade (Figure 1D). To investigate the panicle blast resistance of the population, 249 cultivars were inoculated with the blast strain Hoku1 at the booting stage in the field (Nanjing, China) over different years. The percentage of diseased grains varied greatly from 0 to 100%, indicating a broad range of panicle blast resistance levels (Figure 2A,B). The average percentage of diseased grains in the GJ subgroups was 19% and 22% in 2020 and 2021, respectively, whereas, in the XI subgroups, it was 9% and 8%, showing a higher susceptibility compared to the XI subgroups (Figure 2C,D). A moderate positive linear correlation (*r* = 0.62, *p*-value < 2.2 × 10^−16^) was calculated between phenotypes in two different years, indicating that the data can be treated as replicated data for GWAS analysis.

### 2.2. Identification of Blast-Resistant Loci in Whole Genome

A genome-wide association study (GWAS) was conducted to identify blast-resistant loci (BRL) using a mixed linear model (MLM) based on 3,883,371 high-quality SNPs (MAF > 0.05, missing rate < 50%). The kinship and population structure were used as covariates to control for false positives; the QQ plot still exhibits inflation, perhaps because disease resistance is a result of polygenicity. In 2020, a total of 2873 SNPs (−log10*p* > 4) showed a significant association with the blast resistance trait, and 3118 SNPs were significantly associated in 2021. Among them, 727 SNPs were detected in both years. Based on the significant associated SNPs, 37 and 38 BRLs were identified in the two respective years. In 2020, the identified 37 loci were distributed on all 12 chromosomes. In 2021, no loci were identified on chromosomes 8 and 9 (Figure 3A,D). Out of these loci, 10 BRLs had been previously reported, and 7 BRLs were consistently identified in both years. Specifically, BRL3, BRL5, BRL8, and BRL11 were co-localized with known blast resistance genes *Pi63*, *Pi56*, *pb1*, and *Pita*, respectively [3,27,28,29] (Table 1). BRL1, BRL14, BRL15, BRL19, and BRL20 have been reported by Li et al. (2019) [30], and BRL17 has been reported by Liu et al. (2020) [25]. BRL2 and BRL13, BRL4 and BRL17, BRL6 and BRL18, BRL7 and BRL19, BRL9 and BRL21, BRL10 and BRL22, as well as BRL12 and BRL23, were identified repeatedly in 2020 and 2021 (Table 1).

### 2.3. Analysis of the Candidate Genes in BRL10 and BRL22

To further ascertain the resistance-related genes, we analyzed the loci consistently identified in both years. BRL10 and BRL22, located at the end of the long arm of chromosome 11, emerged as repeat loci in both years and were specifically focused on in this study. The combined positions of BRL10 and BRL22, spanning from 28018096 to 28187112, were selected for subsequent pairwise linkage disequilibrium (LD) analysis. An LD block with D’ > 0.85, extending from position 28136278 to 28187112, was chosen as the candidate region (Figure 4A). According to the Rice Genome Annotation Project (http://rice.uga.edu/, accessed on 1 June 2020), this genomic region contains four candidate genes, including LOC_Os11g46870, LOC_Os11g46880, LOC_Os11g46900, and LOC_Os11g46890 (Figure 4A, Table 2). LOC_Os11g46870, LOC_Os11g46880, and LOC_Os11g46900 encode kinase proteins, while LOC_Os11g46890 encodes a protein with an unknown function (Figure 4A, Table 2). Within this region, seven associated SNPs were detected, including six SNPs in the intergenic region and one nonsynonymous SNP in LOC_Os11g46890, causing a Glycine (Gly) to Serine (Ser) substitution at position 183 (Table 3). Because few SNPs are located in the genes, it is challenging to determine disease-resistant casual genes through haplotype analysis. To further investigate the potential resistance genes in this locus, we examined the inducing expression patterns of four candidate genes challenged with the rice blast pathogen. The expression of LOC_Os11g46890 was not detected. Among the remaining three genes, LOC_Os11g46880 showed a significant induced expression pattern in response to the rice blast pathogen but not the other two genes (Figure 4B–D). The gradual increase in the expression level of this gene indicated a positive response to rice blast fungus. Based on this observation, we consider LOC_Os11g46880 as the potential disease-resistant-related gene in this locus.

### 2.4. Pb4 Positively Regulates Resistance to Rice Blast

LOC_Os11g46880 is predicted to contain an extracellular N-terminal signal peptide, GUB domain, EGF domain, EGF-Ca^2+^ domain, and intracellular Ser/Thr protein kinase domain, indicative of a typical WAK protein. There are 125 and 25 WAKs that have been annotated in rice and *Arabidopsis thaliana*, respectively. In rice, the 125 WAKs can be classified into 67 WAK-RLKs containing both extracellular EGF-like domains and the cytoplasmic kinase domain, 28 WAK-RLCKs containing only the cytoplasmic kinase domain, 13 WAK-RLPs containing only the extracellular EGF-like domain, 12 WAK-short genes, which share 40% identity with a longer WAK protein but lack a domain, and 5 WAK-pseudogenes with stop codons or frameshifts in the coding region. The kinase domain of the 67 WAK-RLKs and 27 WAK-RLCKs in rice, along with the 24 *Arabidopsis thaliana* WAKs and LOC_Os11g46880, were used to conduct an ML evolutionary tree. The evolutionary tree revealed that most WAKs were species-specific, and genes in the WAK-RLK or WAK-RLCK were more formed with a clade, indicating a significant correlation between the kinase domain and the extracellular domain. Notably, LOC_Os11g46880 was most similar to the WAK-RLK gene *OsWAK122* in a rice WAK-RLK clade (Figure 5A). To further investigate the role of LOC_Os11g46880 in the infection process of rice blast fungus, we generated overexpression transgenic plants of LOC_Os11g46880. Transgenic lines 22, 23, and 25, which had a higher expression, exhibited significantly enhanced resistance to rice blast compared to the wild type (Figure 5B–D). It was indicated that LOC_Os11g46880 positively regulated resistance to rice panicle blast, and we named it Pb4, following the panicle resistance gene Pb3. Given that Pb4 is a WAK-RLK and most RLKs are located in the cell membrane, we conducted subcellular localization studies of *N. benthamiana* and confirmed its subcellular location in the cell membrane (Figure 6C). Previous reports showed that OsWAK’s expression is under the control of CEBiP, and the extracellular domain of CEBiP can interact with the extracellular domain of CERK [31,32]. In yeast two-hybrid assays, we found that the extracellular domain of Pb4 also interacted with the extracellular domain of CEBiP. However, the full-length Pb4 did not interact with the full-length CEBiP in yeast, which might be due to the presence of a transmembrane domain (Figure 6A,B). Given that CEBiP acts as a receptor for chitin in rice, we hypothesize that the expression of *Pb4* may also be induced by chitin. By treating rice seedlings with chitin, the expression of *Pb4* was indeed induced (Figure 5E). Several WAKs have been reported to act as receptors for OGs. We then treated rice with OGs and found that the expression of *Pb4* was significantly induced (Figure 5F). Taken together, as a WAK-RLK, *Pb4* positively regulates rice resistance against rice panicle blast. The interaction between the extracellular domain of PB4 and CEBiP, as well as the induced expression by chitin and polygalacturonic acid, suggest that *Pb4* may participate in the immune response of rice to chitin and oligogalacturonides.

## 3. Discussion

Rice blast disease, caused by the fungus *Magnaporthe oryzae*, is one of the most devastating diseases that significantly impact rice production globally. Among various disease symptoms, the seedling blast and panicle blast are the most common; however, the economic and production losses caused by panicle blast are more significant [33]. Though more than 100 genes have been identified for seedling blast, very few genes conferring resistance to panicle blast were cloned [33]. Developing and utilizing resistant rice varieties is the most efficient and cost-effective approach to control blast. However, currently, only a few of the cloned R genes can be applied in breeding due to a lack of broad-spectrum and durable resistance. Among the cloned R genes, *Pi1*, *Pi5*, *Pi33*, *Pi54*, *Piz*, *Piz-t*, *Pi2*, *Pi9*, *Pi40,* and *Pigm* confer broad-spectrum resistance to leaf blast [34,35,36,37,38,39,40,41]. *Pb1*, *Pb2*, *Pb3*, *Pi25*, *Pi64*, *Pi68(t)*, and *Pigm* loci confer resistance to panicle blast, and only *Pigm* provides broad-spectrum resistance to panicle blast [3,4,5,6,7,8,42].

Compared to traditional linkage mapping, GWAS is time-saving and labor-saving and can be used to discover novel disease-associated-resistant genes with higher resolution [23]. In this research, we identified 68 blast resistance loci through GWAS assays using 249 rice varieties selected from the 3K rice population. Through the LD block and expression analysis of the candidate genes after inoculating with rice blast fungus, we identified LOC_Os11g46880 as a resistance-related gene for further research and named it *Pb4*. *Pb4* encodes a typical WAK-RLK protein with an N-terminal signal peptide, extracellular GUB domain, EGF domain, EGF-Ca^2+^ domain, and an intracellular Ser/Thr protein kinase domain. The overexpression transgenic plants of *Pb4* showed enhanced resistance to panicle blast with a reduced diseased grains rate compared to the wild type.

As typical RLK proteins, WAKs are a superfamily with 125 annotated genes in rice and 25 genes in *Arabidopsis thaliana*. According to the conserved domains, the 125 WAKs in rice have been classified into five subfamilies, including 67 WAK-RLKs, 28 WAK-RLCKs, 13 WAK-RLPs, 12 WAK-short genes, and 5 WAK-pseudogenes. Previous reports suggest that WAKs are involved in the recognition of carbohydrate-based DAMPs or PAMPs as PRRs [43] and are connected with the cell wall through pectin binding sites [44]. In *Arabidopsis thaliana*, both AtWAK1 and AtWAK2 can bind to pectin through their extracellular non-EGF domains in vitro [45,46]. However, so far, only AtWAK1 has been confirmed as a receptor of oligogalacturonides (OGs) using domain-swapping methods in vivo, which activates the intracellular kinase domain and triggers downstream defense responses [46]. On the other hand, AtWAK2 is required for the activation of numerous genes, including MAPK6 phosphorylation, in the protoplasts upon pectin induction [45]. In this study, the expression of *Pb4* was significantly induced by polygalacturonic acid, suggesting it may be involved in OGs recognition as a PRR protein.

Several studies have directly demonstrated the roles of WAKs in the resistance responses against bacterial and fungal pathogens. In *Arabidopsis*, overexpression of *AtWAK1* enhances resistance to *Botrytis cinerea* [47], while *AtWAKL22* confers dominant resistance to Fusarium wilt disease [48]. In rice, the recently cloned disease resistance gene *Xa4*, encoding a WAK protein, can promote cellulose synthesis, inhibit cell wall loosening, strengthen the plant cell wall, and provide durable resistance against rice bacterial blight [49]. Subcellular localization and apoplastic fractionation experiments in *N. benthamiana* revealed that OsWAK1 is associated with the cell wall, and its expression is induced by the rice blast fungus, salicylic acid (SA), methyl jasmonate (MeJA), and mechanical injury. *OsWAK14*, *OsWAK91*, and *OsWAK92* mediate rice blast resistance, triggering ROS burst and the expression of defense-related genes. As a large gene family in rice, WAKs may form a heteromeric complex to conduct function. Yeast two-hybrid experiments have shown that OsWAK14, OsWAK91, and OsWAK92 can indeed form both homomeric and heteromeric complexes [50,51]. In cotton, GhWAK7A can interact with GhLYK5 and GhCERK1 in the uninfected state. However, during fungal pathogen infection, GhLYK5 can recognize chitin, which is released from fungal cell walls, and interact with GhCERK1 to promote the phosphorylation of GhWAK7A. Phosphorylated GhLYK5 may facilitate and maintain the formation of the GhLYK5-GhCERK1 complex, thus further activating cytoplasmic signaling and eliciting defense responses [16]. We also found that the expression of *Pb4* is inducible by chitin, and the extracellular domain of Pb4 can interact with the extracellular domain of CEBiP in yeast two-hybrid assays. As an RLP protein without an intracellular kinase domain, CEBiP often interacts with an RLK protein, such as CERK1, to transmit signals from the extracellular to intracellular, leading to the activation of downstream signaling pathways. The interaction between Pb4 and CEBiP may indicate they form complexes to recognize external signals and activate intracellular defense signaling.

In summary, we identified a novel non-NLR panicle resistance gene through GWAS, which belongs to the WAK gene family. As an RLK protein, Pb4 may act as a receptor for the OGs derived from the cell wall and form a co-receptor with CEBiP to recognize the chitin released from the *M. oryzae* cell wall to activate PTI. It might also confer broad-spectrum resistance to pathogens through strengthening the plant’s cell wall. The specific mechanisms underlying intracellular signal transmission, as well as the potential recognition of other PAMPs or DAMPs by WAKs and the resistance spectrum of Pb4, require further investigation. In summary, this study identified a novel panicle blast-resistant gene for breeding and provides a theoretical basis for understanding the role of WAKs in mediating rice resistance against rice blast disease.

## 4. Materials and Methods

### 4.1. Plant and Fungal Materials

A total of 249 cultivars from the 3K rice accessions with a relatively uniform growth period were selected to grow in Nanjing, Jiangsu province, which were provided by the Shanghai Institute of Plant Physiology and Ecology. The rice blast strain Hoku1 used in this study was provided by the Institute of Crop Science, Chinese Academy of Agricultural Sciences. The blast fungi were grown on corn–rice–straw agar plates at 28 °C for seven days and then transferred in black light (20 W) in an incubator at 28 °C for sporulation and culture for 7–10 days to promote spores production. For panicle blast inoculation, rice plants in the booting stage were inoculated with the conidial suspension by using an injection method as described previously [52]. Three plants of each line and three panicles per plant were inoculated with a conidial suspension (1 × 10^5^ conidia/mL). The disease severity was visually assessed, and the percentage of diseased grains was determined two weeks after inoculation according to the method previously described [53].

### 4.2. Population Structure and GWAS Analysis

The population structure of the 249 cultivars was analyzed using ADMIXTURE (version 1.3.0) on the core SNPs [54]. The k (number of groups) was set from 2 to 12, and k = 7 was chosen due to its cross-validation error being minimum. The group membership for each sample was defined by applying the threshold of ≥0.65 to this matrix. If the sum of the components for the subpopulations within the major groups XI and GJ was ≥0.65, the samples were classified as XI-adm or GJ-adm, respectively, and the remaining samples were deemed admixed (admix). The construction of phylogenetic trees was performed using FastTree based on the maximum likelihood method using the core SNPs [55].

GWAS analysis of panicle blast resistance was conducted using the mixed linear model in Tassel 5.0 and the filtered SNPs (MAF > 0.05, missing rate < 50%) [54]. The *p*-value < 10^−4^ was chosen as the threshold to define significant SNPs, and a region with more than eight consecutive SNPs (the distance between SNPs was less than 200 kb) was named as a locus. The kinship and population structure (k = 7) were used as covariates. Manhattan and QQ plots were plotted using the R package “cmplot” [56]. A linkage disequilibrium (LD) heatmap was generated with LDBlockShow [57].

### 4.3. RNA Extraction and Real-Time PCR

For the expression pattern analysis of *Pb4*, samples were collected at different time points after the rice blast pathogen inoculation or treated with chitin at a concentration of 10 μg/mL and polygalacturonic acid at a concentration of 100 μg/mL, immediately frozen in liquid nitrogen, and stored at −80 °C. Total RNA was extracted using the ATGPure^®^ Cell/Tissue RNA Extraction Kit (Code: R201). The first-strand cDNA synthesis was then conducted using HiScript^®^ II RT SuperMix (Code: R223-01). For the qRT-PCR, the AceQ^®^ qPCR SYBR Green Master Mix (Product Code: Q111-02) was employed, and the entire qRT-PCR process was carried out on an LC480 II qPCR system (Roche fluorescence quantitative LightCycler480, San Francisco, CA, USA). The expression level of *Actin* was used as an internal control.

### 4.4. Construction of Transgenic Rice Plants

Full-length cDNA of *Pb4* was inserted into the pCAMBIA1300s vector to generate the overexpression transgenic vector driven by the 35s promoter. The verified vector plasmid was then transformed into susceptible rice plants, Suyunuo, using the Agrobacterium-mediated method. The primers YZ-Pb4-F/R and q-Pb4-F/R were used to validate positive transgenic plants and analyze the expression levels of *Pb4* (Appendix A).

### 4.5. Subcellular Localization

Full-length cDNA of *Pb4*, fused with GFP, was constructed into the PCAMBIA1300s-GFP vector driven by the 35s promoter to generate the subcellular localization vector. The Agrobacterium carrying the Pb4-GFP and P19 plasmid were respectively prepared in the infiltration medium (10 mM MgCl_2_, 10 mM MES, 100 μM Acetosyringone, pH = 5.6) to an OD_600_ = 1.0. They were mixed in equal amounts and injected into *N. benthamiana* leaves. After 48 h, the samples were observed using a Leica laser scanning confocal microscope.

### 4.6. Y2H Assay

Y2H assays were conducted using the Gold Yeast Two-Hybrid System (Clontech) protocol. Different lengths of *Pb4* and *CEBiP* cDNA fragments were cloned into the pGADT7 and pGBKT7 vectors, respectively. pGBKT7 and pGADT7 vectors were co-transformed into the Y2H Gold yeast strain. The transformed yeast cells were grown on selection plates (SD/-Leu/-Trp/-His/-Ade) to detect the interactions.

## Figures and Tables

**Figure 1 ijms-25-00830-f001:**
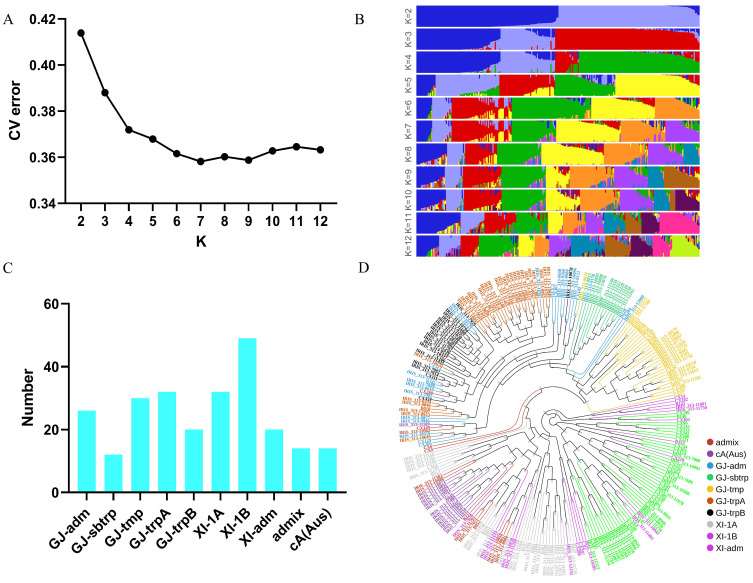
Population structure of the 249 rice accessions. (**A**) Plot of the CV error with k ranging from 2 to 12. The k means the subgroups of the populations. (**B**) Population structure based on different ks. Each color represents a different subgroup. (**C**) The number of rice accessions in different subgroups based on K = 7. GJ-sbtrp = Geng/japonica subtropical subpopulation, GJ-trp = Geng/japonica tropical subpopulation, GJ-tmp = Geng/japonica temperate subpopulation, GJ-adm = Geng/japonica admixed types between two or more GJ subpopulations, XI-1 = Xian/indica subpopulation 1, XI-adm = Xian/indica admixed types between two or more XI subpopulations, cA (Aus) = centrum-Aus population, admix = admixed between any two or more of the XI, GJ, or cA (Aus) populations. (**D**) Phylogenetic tree of the 249 rice accessions; samples are colored by their assignment to k = 7 subgroups from Admixture.

**Figure 2 ijms-25-00830-f002:**
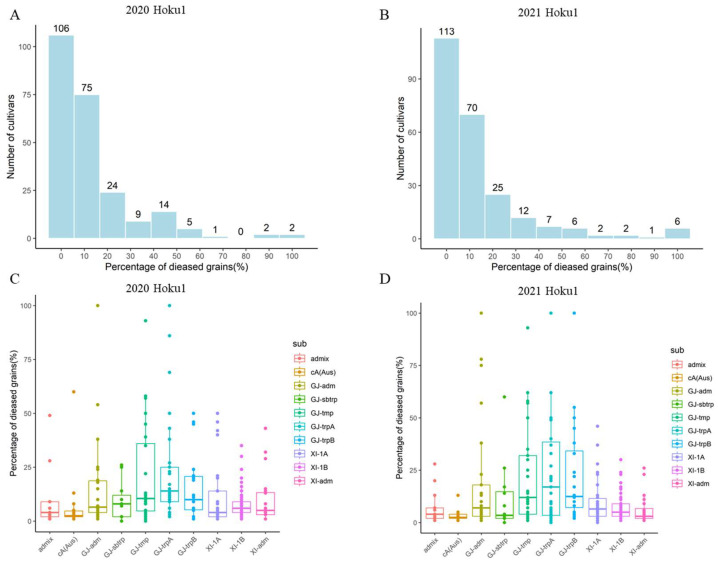
The panicle blast resistance of the 249 rice accessions. (**A**,**B**) The frequency distribution of panicle blast phenotypes for 249 rice varieties inoculated with the Hoku1 strain in the years 2020 and 2021, respectively. (**C**,**D**) Box plots for panicle blast resistance, based on the K = 7 subgroups. The horizontal line in the center of each box denotes the median. The upper and lower limits of each box represent quartiles.

**Figure 3 ijms-25-00830-f003:**
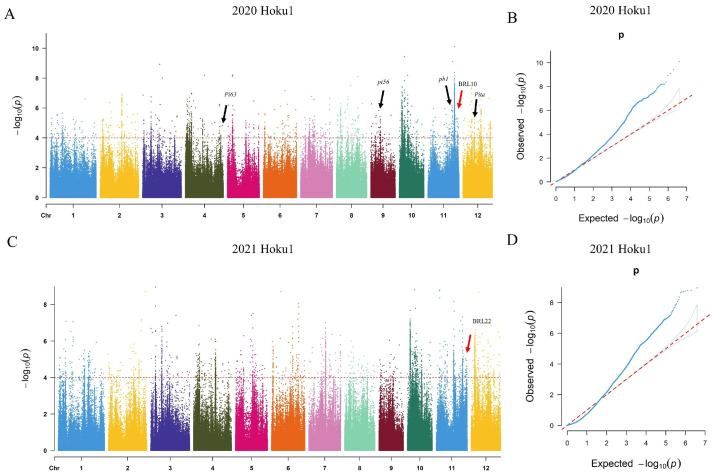
GWAS of the 249 rice accessions for identification of the rice blast-resistant candidate genes. (**A**,**C**) Manhattan plot of the SNPs on 12 rice chromosomes. The *x*-axis represents the genomic position of the SNPs along each chromosome; the *y*-axis indicates the transformed *p*-value, −log10*p*. The black arrows indicate the cloned R genes, while the red arrows indicate the repeat loci BRL10 and BRL22 newly identified in this study. (**B**,**D**) Quantile–Quantile plots of expected and observed *p*-value, −log10*p*.

**Figure 4 ijms-25-00830-f004:**
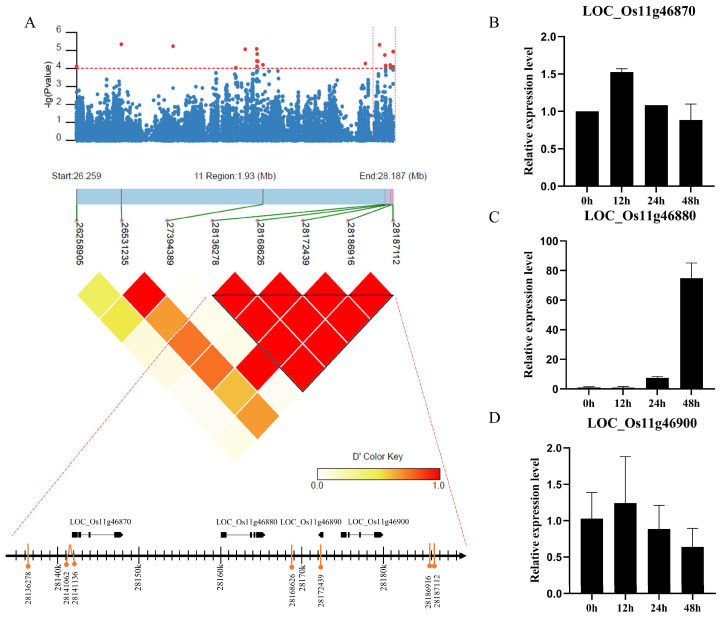
Identification of the rice blast-resistant candidate genes. (**A**) Local Manhattan plot (**top**) and LD heatmap (**bottom**) on BRL10 (**left**). Candidate genes and significant SNPs in the selected LD block (**right**). (**B**–**D**) qRT-PCR analysis of the candidate genes during *M. oryzae* infection. The *x*-axis indicates the time points post-inoculation. The *y*-axis indicates the relative transcription level of candidate genes. Error bars, mean ± SD (*n* = 3).

**Figure 5 ijms-25-00830-f005:**
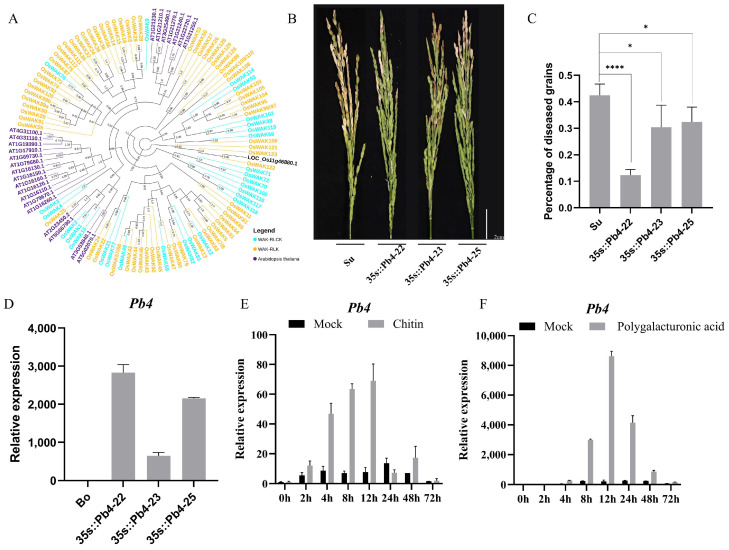
*Pb4* positively regulates resistance to rice blast. (**A**) Evolutionary tree of Pb4, along with 67 WAK-RLKs, 27 WAK-RLCKs (1 WAK-RLCK was not predicted a kinase domain) in rice, and 24 WAKs (1 WAK was not predicted a kinase domain) with kinase domain in *Arabidopsis*. The purple branches represent *Arabidopsis* WAKs, the yellow branches represent rice WAK-RLKs, the blue branches represent rice WAK-RLCKs, and the black branch corresponds to LOC_Os11g46880, which is Pb4. (**B**,**C**) Increased blast resistance of *Pb4*. Panicles were inoculated with *M. oryzae* at the booting stage. The diseased grains were counted at 10 dpi. Scale bar, 2 cm. * *p* < 0.05, **** *p* < 0.0001. (**D**) Expression levels of *Pb4* in overexpression plants. Error bars, mean ± SD (*n* = 3). (**E**,**F**) Two-week-old rice seedlings were treated with chitin (10 μg/mL) or polygalacturonic acid (100 μg/mL). Error bars indicate mean ± standard deviation (*n* = 3). Two-week-old rice was treated with chitin or polygalacturonic acid. Error bars, mean ± SD (*n* = 3).

**Figure 6 ijms-25-00830-f006:**
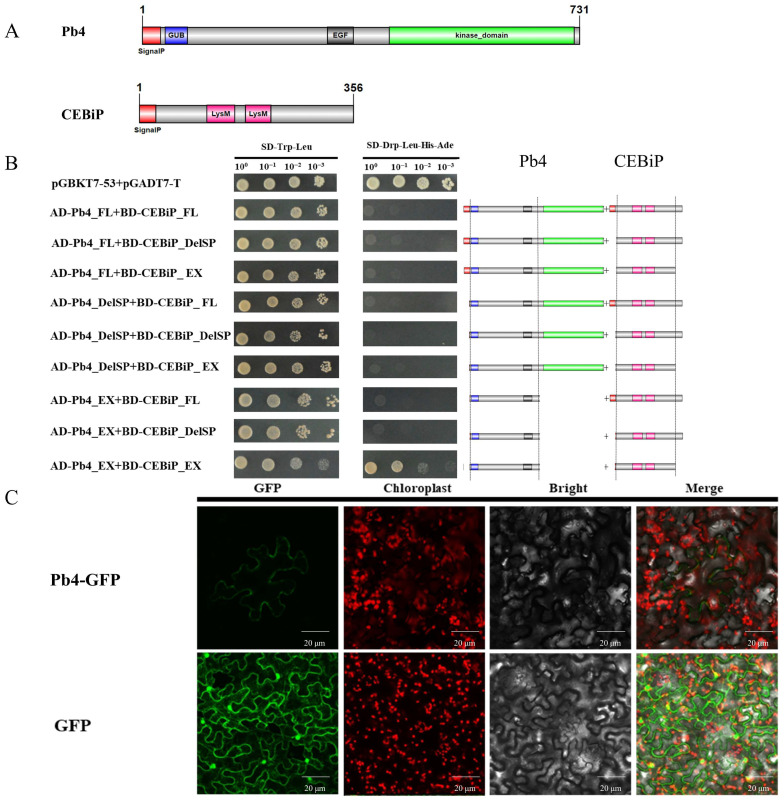
Pb4 interacted with CEBiP and was located in the cell membrane. (**A**) Protein structure diagram of Pb4 and CEBiP. (**B**) Interaction between the extracellular domains of Pb4 and CEBiP in yeast. The right shows the protein structure diagrams of truncated Pb4 and CEBiP. FL: full length; DelSP: Delete Signal Peptide; EX: extracellular (excluding SP); CY: cytoplasmic. (**C**) Subcellular localization of Pb4-GFP on the cell membrane in tobacco cells.

**Table 1 ijms-25-00830-t001:** Part of the loci identified in 249 rice varieties in 2020 and 2021 by inoculating with the Hoku1.

Locus	Chr	Position	Top SNP	*p*-Value	Years	Locus Reference
BRL1	3	15255983-15434525	118389	4.86 × 10^−5^	2020	[30]
BRL2	4	4088528-4505882	4088552	3.4708 × 10^−6^	2020	
BRL3	4	32972226-33188952	33188952	1.626 × 10^−5^	2020	*Pi63*
BRL4	6	1454195-1604484	1490561	2.461 × 10^−6^	2020	
BRL5	9	8674158-9304868	9069693	1.812 × 10^−5^	2020	*pi56*
BRL6	10	4221890-5647528	4798719	1.5076 × 10^−8^	2020	
BRL7	10	7667073-8691546	7837737	6.7119 × 10^−9^	2020	
BRL8	11	22734774-23104761	22905810	9.0682 × 10^−7^	2020	*pb1*
BRL9	11	24235297-25570330	24926857	7.8047 × 10^−11^	2020	
BRL10	11	28018096-28187112	28103725	4.9804 × 10^−6^	2020	
BRL11	12	10653088-10797158	10734066	9.7844 × 10^−6^	2020	*Pita*
BRL12	12	15203722-15771634	15673466	6.6541 × 10^−7^	2020	
BRL13	4	4112569-4517761	4444275	1.648 × 10^−6^	2021	
BRL14	4	5459995-5745230	5590542	1.4732 × 10^−6^	2021	[30]
BRL15	5	7403759-7735177	7735177	5.0795 × 10^−7^	2021	[30]
BRL16	5	16077078-16438269	16438239	3.0808 × 10^−8^	2021	[25]
BRL17	6	1216276-1514569	1490561	2.0864 × 10^−6^	2021	
BRL18	10	4203994-5367946	4749066	5.9358 × 10^−7^	2021	
BRL19	10	6200492-9329504	6525986	1.571 × 10^−9^	2021	[30]
BRL20	11	8017796-8518194	8409165	1.5501 × 10^−6^	2021	[30]
BRL21	11	24202727-24319531	24319531	1.3067 × 10^−7^	2021	
BRL22	11	28136278-28187112	28186726	1.203 × 10^−5^	2021	
BRL23	12	15535945-16059900	15673466	1.0786 × 10^−7^	2021	

**Table 2 ijms-25-00830-t002:** The annotation of the candidate genes.

Gene	Annotation	Position	Length
LOC_Os11g46880	protein kinase domain containing	Chr11:28159898-28165301	711
LOC_Os11g46870	protein kinase	Chr11:28142191-28148337	680
LOC_Os11g46900	wall-associated receptor kinase	Chr11:28174980-28179995	707
LOC_Os11g46890	expressed protein	Chr11:28172388-28172985	199

**Table 3 ijms-25-00830-t003:** Types of variations observed in significant SNPs within the candidate region.

SNP	Ref	Alt	Mutant	Gene	Change
Chr11-28136278	G	A	Intergenic	LOC_Os11g46860-LOC_Os11g46870	-
Chr11-28141062	C	T	Intergenic	LOC_Os11g46860-LOC_Os11g46870	-
Chr11-28141136	G	A	Intergenic	LOC_Os11g46860-LOC_Os11g46870	-
Chr11-28168626	G	A	Intergenic	LOC_Os11g46880-LOC_Os11g46890	-
Chr11-28172439	C	T	Nonsynonymous	LOC_Os11g46890	Gly183Ser
Chr11-28186916	G	T	Intergenic	LOC_Os11g46900-LOC_Os11g46910	-
Chr11-28187112	C	A	Intergenic	LOC_Os11g46900-LOC_Os11g46910	-

## Data Availability

Data are contained within the article and Appendix A.

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
