# Peer review of "Genome-Wide Association Study Identifies Rice Panicle Blast-Resistant Gene Pb4 Encoding a Wall-Associated Kinase"

_ijms, 2024, doi:10.3390/ijms25020830_

Round 1

Reviewer 1 Report

Comments and Suggestions for Authors

Fan et al. identified multiple loci that potentially confer resistance to rice panicle blast, including Pb4. It is well-written and presented nicely. I suggest minor revisions for the manuscript.

1. Sometimes, gene names are italics but other times, they are not. It would be better to be consistent. 

2. Sometimes, full names for abbreviations were missing when they were initially introduced. Such as DAMPs in line 50.

3. Line 106- Pearson correlation- p-value is missing.

4. Phylogenetic trees in the manuscript might need to be high resolution. Another critical point is they don't show posterior probability values that support branch construction. 

5. BRL10 & BRL22 can be indicated in Figure 3 A&C. Also, the author should mention the threshold for the Manhattan plots. 

6. Table 1 p-value... "p" should be italic

7. Line188-194- might have to check with the sentence structure and grammar.

Again, these are recommendations that the authors need a bit of modification. 

Comments on the Quality of English Language

Minor errors

Author Response

Response to reviewers’ comments

Reviewer 1:

Fan et al. identified multiple loci that potentially confer resistance to rice panicle blast, including Pb4. It is well-written and presented nicely. I suggest minor revisions for the manuscript.

  1. Sometimes, gene names are italics but other times, they are not. It would be better to be consistent.

Answer: Thanks for reviewers’ comments. we checked the gene names in the manuscript and make sure they are correct. Some gene names which are not italics represent the proteins according to context of the sentence.

  1. Sometimes, full names for abbreviations were missing when they were initially introduced. Such as DAMPs in line 50.

Answer: Thanks for reviewers’ comments. we checked the abbreviations in the manuscript and added the full name when they are first used.

  1. Line 106- Pearson correlation- p-value is missing.

Answer: Thanks for reviewers’ comments. we added the P value after the pearson correlation in line 114.

  1. Phylogenetic trees in the manuscript might need to be high resolution. Another critical point is they don't show posterior probability values that support branch construction.

Answer: Thanks for reviewers’ comments. We substituted the phylogenetic tree with a higher resolution and added the support value (Figure 1D, Figure 5A). The phylogenetic tree was conducted with fasttree which computes local support values with the Shimodaira-Hasegawa test but not bootstrap because of a large data needed to process, so the support value is ranging from 0 to 1.

  1. BRL10 & BRL22 can be indicated in Figure 3 A&C. Also, the author should mention the threshold for the Manhattan plots.

Answer: Thanks for reviewers’ comments. We added the BRL10 and BRL22 in the Figure 3A, C and the threshold for the Manhattan plots was described in the Materials and Methods sections in line 432.

  1. Table 1 p-value... "p" should be italic

Answer: Thanks for reviewers’ comments. We changed the “p-value” in Table 1 to “P-value”.

  1. Line188-194- might have to check with the sentence structure and grammar.

Answer: Thanks for reviewers’ comments. We checked and corrected the structure and grammar of the sentence in line 227-233 to make the sentence clearer.

Again, these are recommendations that the authors need a bit of modification.

Reviewer 2 Report

Comments and Suggestions for Authors

This manuscript describes identification of new blast resistance gene, Pb4 through genome-wide association study (GWAS). The authors also conducted complementation, expression, and gene interaction analyses and revealed that Pb4 is indeed involved in rice blast resistance. The result itself is interesting and useful for the future rice breeding programs. However, at this stage, the quality of this manuscript is insufficient and should be revised based on the following comments:

(Major points)

1. Based on the q-q plots, the quality of GWAS in this study is not high and the obtained peaks may include many false positives. The authors should discuss the reason and the measure to be taken for this. In addition, why did you set threshold at -log10P = 4?

2. The authors discuss population structure analyses based on phylogenetic and ADMIXTURE analysis, however, in the GWAS, they use PCA results as population structure input. It is strongly recommended that the authors explain the reason for this. If you use the PCs, it is also recommended that plots for PCs you used are visualized and shown in the text. How many principal components did you use?

3. Figure 3: It is recommended that the positions of major locus such as pi63, pi56, pb1, pita etc. are also shown in Manhattan plots.

4. L151: The authors picked up the loci, BRL10 and BRL22, though there were other candidates. As the reasons were insufficiently explained, it feels abrupt.

5. Results of yeast two hybrid experiments should be more explained in the main text. For example, we can understand the interaction between extracellular domains of Pb4 and CEBiP (Fig. 6B, AD-Pb4_EX+BD-CEBiP_EX), however, why interactions weren’t identified between full length sequences, despite they both containing extracellular domains?

6. English language of this manuscript is sometimes inappropriate and should be edited by a native speaker.

(Minor points)

1. Abbreviations are not well defined (for example, NLR(L37), RLK(L44), LRRs(L45), DAMPs(L50) etc.). Please recheck the whole part of your manuscript.

2. L74: SMART seq and Nanopore seq are both included in next generation sequencing, I think.

3. L82: populationscultivars

4. L106: correlation coefficient is not r2 but r.

5. Tables and figures should be mentioned in the order of insertion. Please check the order as sometimes there are errors in the order (for example, L188).

6. L210: Citation is required for CEBiP.

7. Figure5E, F: Colors of bar graphs for MOCK and Chitin are difficult to be distinguished.

8. Discussion: Please remove duplicate content from the Introduction if possible.

Comments on the Quality of English Language

English language of this manuscript is sometimes inappropriate and should be edited by a native speaker.

Author Response

Response to reviewers’ comments

Reviewer 2:

This manuscript describes identification of new blast resistance gene, Pb4 through genome-wide association study (GWAS). The authors also conducted complementation, expression, and gene interaction analyses and revealed that Pb4 is indeed involved in rice blast resistance. The result itself is interesting and useful for the future rice breeding programs. However, at this stage, the quality of this manuscript is insufficient and should be revised based on the following comments:

(Major points)

  1. Based on the q-q plots, the quality of GWAS in this study is not high and the obtained peaks may include many false positives. The authors should discuss the reason and the measure to be taken for this. In addition, why did you set threshold at -log10P = 4?

Answer: Thanks for reviewers’ comments. Regarding the QQ plot quality in our GWAS, we have already added kinship and population structure as covariates to control for false positives (mentioned in line 435). The persistence of this issue may be due to the fact that disease resistance is polygenicity. To further avoid interference from false positive peaks, we selected repeatedly identified loci for analysis. For the reason of setting threshold at -log10P = 4, it can help us to find Micro-effect QTL loci and the threshold is widely used in rice such as Yu et al., 2023, Liu et al., 2020 and Lv et al., 2015.

  1. The authors discuss population structure analyses based on phylogenetic and ADMIXTURE analysis, however, in the GWAS, they use PCA results as population structure input. It is strongly recommended that the authors explain the reason for this. If you use the PCs, it is also recommended that plots for PCs you used are visualized and shown in the text. How many principal components did you use?

Answer: Thanks for reviewers’ comments. We are sorry for the mistake that the “PCA” should be “population structure”. In our MS, the population structure was analyzed by ADMIXTURE. So, the GWAS analysis was also conducted by using the population structure as a covariate. We changed “PCA” to “population structure” in line 435.

  1. Figure 3: It is recommended that the positions of major locus such as Pi63, pi56, pb1, Pita etc. are also shown in Manhattan plots.

Answer: Thanks for reviewers’ comments. We added Pi63, pi56, pb1, Pita, BRL10 and BRL21 loci in the Figure 3A, C.

  1. L151: The authors picked up the loci, BRL10 and BRL22, though there were other candidates. As the reasons were insufficiently explained, it feels abrupt.

Answer: Thanks for reviewers’ comments. The other repeatedly identified loci, such as BRL2 and BRL13, BRL4 and BRL17, BRL6 and BRL18, BRL7 and BRL19, BRL9 and BRL21, BRL12 and BRL23, are also under investigation and the researches on other QTL loci are still ongoing and have not been published yet, thus we provided less analyze on other loci, in this article our main focus was on the analysis of the BRL10 and BRL21 loci.

  1. Results of yeast two hybrid experiments should be more explained in the main text. For example, we can understand the interaction between extracellular domains of Pb4 and CEBiP (Fig. 6B, AD-Pb4_EX+BD-CEBiP_EX), however, why interactions weren’t identified between full length sequences, despite they both containing extracellular domains?

Answer: Thanks for reviewers’ comments. This may be due to the presence of transmembrane domains, which result in different three-dimensional spatial structures between full-length proteins and truncated proteins in yeast and we also added the explanation of this result in main text in line 270. Whether the interaction between full length sequences exists maybe need to be verified in plants.

  1. English language of this manuscript is sometimes inappropriate and should be edited by a native speaker.

Answer: Thanks for reviewers’ comments. We have carefully checked and improved the English writing in the revised manuscript.

(Minor points)

  1. Abbreviations are not well defined (for example, NLR(L37), RLK(L44), LRRs(L45), DAMPs(L50) etc.). Please recheck the whole part of your manuscript.

Answer: Thanks for reviewers’ comments. we have checked the abbreviations in the manuscript and added the full name when they are first used.

  1. L74: SMART seq and Nanopore seq are both included in next generation sequencing, I think.

Answer: Thanks for reviewers’ comments. We changed “next generation sequencing” to “second generation sequence” in line 80.

  1. L82: populations→cultivars

Answer: Thanks for reviewers’ comments. We changed “populations” to “cultivars” in line 87.

  1. L106: correlation coefficient is not r2 but r.

Answer: Thanks for reviewers’ comments. We changed “r2” to “r” in line 114.

  1. Tables and figures should be mentioned in the order of insertion. Please check the order as sometimes there are errors in the order (for example, L188).

Answer: Thanks for reviewers’ comments. We checked the order of the tables and figures and make sure the order is correct.

  1. L210: Citation is required for CEBiP.

Answer: Thanks for reviewers’ comments. We added the citation for CEBiP in line 268.

  1. Figure 5E, F: Colors of bar graphs for MOCK and Chitin are difficult to be distinguished.

Answer: Thanks for reviewers’ comments. We changed the colors of bar for graphs for MOCK and treated groups in Figure 5E, F.

  1. Discussion: Please remove duplicate content from the Introduction if possible.

Answer: Thanks for reviewers’ comments. We removed the duplicate content from introduction in discussion in line 324.

Round 2

Reviewer 2 Report

Comments and Suggestions for Authors

I appreciate for responding to my comments. I agree that the authors addressed most of the comments. However, the quality of English is still inadequate. I think they might have corrected their English by themselves. For example, a newly added sentence, "but a full-length interaction between them was not abserved which maybe because of the presence of a transmembrane domain (Figure 6A, B)." includes gramatical mistakes. Also in other parts, there are some sentences which are redundant and difficult to be understood. I strongly recommemd again that the manuscript is corrected by a native speaker.

(Other minor points)

1. L107-110: r=0.62 is seemingly not high enough to interpret as "strong correlation". Please reconsider the expression.

2. For my comments 1 and 4, your answers should be more included in the main text, as far as its OK with you.

Comments on the Quality of English Language

The quality of English is still inadequate. I think they might have corrected their English by themselves. For example, a newly added sentence, "but a full-length interaction between them was not abserved which maybe because of the presence of a transmembrane domain (Figure 6A, B)." includes gramatical mistakes. Also in other parts, there are some sentences which are redundant and difficult to be understood. I strongly recommemd again that the manuscript is corrected by a native speaker.

Author Response

Response to reviewers’ comments

Reviewer 2:

I appreciate for responding to my comments. I agree that the authors addressed most of the comments. However, the quality of English is still inadequate. I think they might have corrected their English by themselves. For example, a newly added sentence, "but a full-length interaction between them was not observed which maybe because of the presence of a transmembrane domain (Figure 6A, B)." includes grammatical mistakes. Also in other parts, there are some sentences which are redundant and difficult to be understood. I strongly recommend again that the manuscript is corrected by a native speaker.

Answer: Thanks for reviewers’ comments. We further refined the grammar and structure of sentences in this manuscript, making the expression clearer, more concise, and enhancing overall readability and comprehensibility. We hope that this revised manuscript meets the requirements.

(Other minor points)

  1. L107-110: r=0.62 is seemingly not high enough to interpret as "strong correlation". Please reconsider the expression.

Answer: Thanks for reviewers’ comments. We modified the expression 'strong positive linear correlation' to 'moderate positive linear correlation' in line 107.

  1. For my comments 1 and 4, your answers should be more included in the main text, as far as its OK with you.

Answer: Thanks for reviewers’ comments. We have added more detailed explanations in lines 132-134 and lines 157-159 to make the article more readable and understandable, based on comments 1 and 4.